# Hereditary Aortopathies as Cause of Sudden Cardiac Death in the Young: State-of-the-Art Review in Molecular Medicine

**DOI:** 10.3390/diseases12110264

**Published:** 2024-10-23

**Authors:** Cecilia Salzillo, Andrea Marzullo

**Affiliations:** 1Department of Experimental Medicine, PhD Course in Public Health, University of Campania “Luigi Vanvitelli”, 81100 Naples, Italy; 2Department of Precision and Regenerative Medicine and Ionian Area, Section of Pathology, University of Bari “Aldo Moro”, 70121 Bari, Italy

**Keywords:** aortopathies, hereditary aortopathies, sudden cardiac death, young, genetics, biomedicine, molecular medicine, molecular autopsy

## Abstract

Hereditary aortopathies are a group of rare genetic diseases affecting the aorta and its major branches, and they represent a cause of sudden cardiac death. These pathologies are classified into syndromic hereditary aortopathies and non-syndromic hereditary aortopathies. The epidemiology of hereditary aortopathies varies according to the specific genetic condition involved; however, these disorders are believed to account for a significant proportion of sudden cardiac death in young individuals with a family history of inherited cardiovascular conditions. The causes of hereditary aortopathies are primarily genetic, with pathogenic variants in various genes encoding structural proteins of the vascular wall, leading to dissection, aneurysms, rupture, and ultimately sudden cardiac death. When the cause of death remains unknown after an autopsy, it is referred to as sudden unexplained death, and post-mortem genetic testing, known as a molecular autopsy, is crucial to confirm hereditary aortopathies and assess the genetic risk in the patient’s relatives. This helps to facilitate diagnostic and therapeutic pathways and/or implement monitoring strategies to prevent sudden cardiac death. In this state-of-the-art review, we focus on syndromic and non-syndromic hereditary aortopathies causing sudden cardiac death in the young and explore preventive strategies for affected family members.

## 1. Introduction

Aortopathies are a group of acquired or hereditary diseases affecting the aorta, characterised by structural alterations in the aortic wall, predisposing it to dilation and aneurysm formation [1,2]. In young people and children, the most common causes of aortopathy are congenital heart defects and hereditary genetic conditions, leading to the classification of hereditary aortopathies, which are associated with a high risk of aortic dilatation, aneurysm, and dissection at a young age [3,4], potentially resulting in sudden cardiac death (SCD) [5].

Currently, SCD represents a significant international public health issue, accounting for approximately 15–20% of all deaths [6,7], particularly in younger populations. Sudden cardiac death in the young (SCDY) is defined as “SCD affecting individuals under the age of 35, or between 1 and 35 years” [8,9], encompassing both athletes and non-athletes [10,11], with multiple hereditary/genetic or non-hereditary/genetic causes. Specifically, the incidence of SCD in young people (1–35 years) varies according to the age range of the population under study.

In individuals under 30 years of age, the overall risk of SCD is estimated to be approximately 1–2.8 per 100,000 [12,13], with a case rate of 19% in children aged 1–13 years and 30% in adolescents aged 14–21 years [14]. When the cause remains unknown, it is referred to as sudden unexplained death (SUD), which in young people can be caused by undiagnosed cardiomyopathies, channelopathies, and aortopathies. In these instances, a post-mortem genetic study [7], known as molecular autopsy (MA), becomes crucial in initiating a diagnostic–therapeutic and/or monitoring process for the victims’ relatives, thereby reducing the risk of SCD.

In a 2023 study by Wang S. et al. [15], post-mortem genetic analysis was conducted on young adults aged 18–40 using whole-exome sequencing (WES) to identify new genes that predispose to SUD. Of the 27 cases analysed, 51.9% exhibited functionally significant genetic variants in 255 genes already associated with SUD, and 33 new candidate genes were identified, primarily linked to myopathy, which may increase the risk of SUD.

Of the 33 new genes identified, 14—including *RYR1*, *LMO7*, *SYNM*, *LAMA2*, *PLEC*, *SGCA*, *MYBPC2*, *OBSCN*, *ABCB1*, *ABCG2*, *CDH23*, *DYSF*, *NEB*, and *PANK2*—have previously been associated with cardiomyopathies, while the remaining 19—such as *COL12A1*, *COL6A3*, *CYP3A5*, *DPYS*, *EVC2*, *FAT1*, *FLNB*, *HSPG2*, *LAMB2*, *LMCD1*, *MYH3*, *MYOG*, *NUP98*, *PDE8B*, *PERM1*, *PKD1L2*, *RYR3*, *SPTB*, and *THBS1*—show potential correlations.

The inclusion of these new genes improved the genetic diagnostic yield from 51.9% to 66.7%, suggesting their potential for inclusion in future genetic testing for SUD.

## 2. Hereditary Aortopathies

### 2.1. Pathogenesis

Hereditary aortopathies (HAs) are a group of genetic diseases that affect the structure and function of the aorta.

The etiopathogenesis of HA is closely linked to pathogenic variants in the genes that code for structural or regulatory proteins involved in the architecture and function of the extracellular matrix and smooth muscle of the aortic wall. These pathogenic variants alter the composition of the connective tissue of the aortic wall, influencing the elastic components and collagen fibres, and cause a weakening of the wall, with consequent predisposition to the development of aneurysms, dissections, and other complications.

The main pathogenetic mechanisms involve the proteins of the extracellular matrix, the TGFB signalling pathway, and the proteins of the cytoskeletal contraction apparatus [16].

Alterations of the extracellular matrix caused by pathogenic variants in the *FBN1* (fibrillin-1) and *COL3A1* (collagen 3 α-1) genes compromise the structure of the extracellular matrix, resulting in fragility and loss of elasticity of the aortic wall. These pathogenic variants are respectively responsible for Marfan syndrome and vascular Ehlers–Danlos syndrome [16,17,18,19].

Deregulation of TGF-β signalling caused by pathogenic variants in the *TGFBR1* (TGFβ-1 receptor), *TGFBR2* (TGFβ-2 receptor), and *TGFB2* (TGFβ-2) genes alter transforming growth factor beta (TGF-β) signalling, a key regulator of the balance between synthesis and degradation of the extracellular matrix, resulting in weakening of the aortic wall. These pathogenic variants are responsible for Loeys–Dietz syndrome [16,17,18,19].

Smooth muscle cell dysfunction caused by pathogenic variants in the *ACTA2* (α-smooth muscle actin) and *MYH11* (smooth muscle myosin) genes alter the contractile function of aortic wall smooth muscle cells, contributing to aneurysm susceptibility. These pathogenic variants are responsible for different forms of familial aortic aneurysm [16,17,18,19].

Other proteins with pathogenic variants in *TACCA1* and *KCNJ2* genes are responsible for bicuspid aortic valve [16].

### 2.2. Histopathology

In 2016, the Society for Cardiovascular Pathology (SCVP) and the Association For European Cardiovascular Pathology (AECVP) published a document “Consensus statement on surgical pathology of the aorta from the Society for Cardiovascular Pathology and the Association For European Cardiovascular Pathology: II Noninflammatory degenerative diseases—nomenclature and diagnostic criteria”, which represents a terminological consensus to describe degenerative histopathology of the aorta, aimed at standardizing terms to facilitate scientific and diagnostic communication [20].

The main term is “medial degeneration”, a complex form of damage that includes several types of lesions:Accumulation of mucoid extracellular matrix (MEMA);Fragmentation and/or loss of elastic fibres;Thinning of elastic fibres;Disorganization of elastic fibres;Loss of smooth muscle cell nuclei;Medial laminar collapse;Disorganization of smooth muscle cells;Medial fibrosis.

This degeneration can be graded as mild, moderate, or severe.

The main component is MEMA, which can be intralamellar when it does not alter the structure of the lamellae or translamellar when it causes alteration of the lamellae, and the severity is evaluated according to the amount and distribution of the accumulated material.

Another common feature is the fragmentation and loss of elastic fibres, graded based on severity and their diffusion along the aortic wall.

Thinning and disorganization of elastic fibres would indicate a loss of integrity of the elastic structure of the aortic wall.

Loss of smooth muscle cell nuclei is indicative of severe cell damage and may appear patchily or with continuous bands along the aortic wall.

Medial laminar collapse is a form of degeneration in which the elastic lamellae become compacted, thin, or dense and lose function; this feature is assessed based on distribution.

Medial fibrosis can be intralamellar if it does not alter the elastic lamellae or translamellar with alteration of their structure and is evaluated according to its extension.

Overall, the classification of degenerative aortic lesions is based on well-defined criteria of severity and distribution, useful for a precise diagnosis and for monitoring the evolution of aortic pathologies.

Instead, HAs share some histopathological features, especially the fragmentation and/or loss of elastic fibres, but at present, it is not yet possible to differentiate them based on particular morphological patterns; therefore, the association with clinical and genetic features is essential for diagnosis.

### 2.3. Classification

HAs can be classified into syndromic hereditary aortopathies (SHAs) based on the specific genetic syndrome or into non-syndromic hereditary aortopathies (N-SHAs) based on the causative mutation (Table 1).

## 3. Syndromic Hereditary Aortopathies

### 3.1. Marfan Syndrome

Marfan syndrome (MFS) is the most common hereditary connective tissue disease belonging to the group of collagen diseases, which is associated with a reduction in life expectancy due to aortic complications such as dilation and reduction in the aortic root [21] up to sudden cardiac death in young people [22].

Clinically, MFS manifests itself with a broad spectrum of signs and symptoms that affect different organs and systems, mainly the cardiovascular, ocular, and musculoskeletal systems, but it can also affect the lung, skin, and central nervous system. Clinical manifestations vary in severity from mild to the presentation of a severe and rapidly progressive disease [23].

Cardiovascular manifestations affect the valves and the aorta.

Mitral valves prolapse with consequent mitral insufficiency is common in MS and, less frequently, the tricuspid valve, with consequent aortic insufficiency [24,25].

One of the main problems in MFS patients is the dilation of the ascending aorta, up to the formation of an aneurysm and, if untreated, aortic dissection, which represents one of the main causes of sudden death [24,25].

The diagnosis of MFS is based on specific clinical criteria called the Ghent criteria that were codified in 1996, and subsequently revised in 2010 [26], according to which the manifestation of both main features such as aortic root involvement and ectopia of the lens are sufficient for a definitive diagnosis.

MFS is one of the most frequent monogenic malformation syndromes, with an incidence of 1:3000–1:5000, without differences between sex or ethnic groups [21], and if untreated, the average life expectancy of patients is about 40 years [27].

MFS is transmitted predominantly in an autosomally dominant manner, with complete penetrance and variable expression, although in twenty-five percent of cases it occurs sporadically due to de novo mutations [21,23].

In 90% of cases, the mutated gene in MFS is *FBN1*, which is a large gene with 65 exons located on chromosome 15q-21.1 that produces fibrillin-1 [28,29], a glycoprotein of the extracellular matrix that is the main constituent of elastic fibres [30]. The most frequent type of mutation in the *FBN1* gene is the point mutation, with 60% of missense mutations and 10% of nonsense mutations [29].

In less than 10% with a typical Marfan phenotype, the mutation in the FBN1 gene is not identifiable due to complete deletion of the allele or altered regulation [21].

Instead, in patients with atypical presentations, the cause may be a mutation in a gene encoding the transforming growth factor-beta receptor (TGFBR), in particular *TGFBR1* or *TGFBR2* [21].

Normally, fibrillin-1 is a regulator of the bioavailability of TGF-beta, a growth factor responsible for many cellular processes, such as inflammation and tissue repair. When the protein is mutated, TGF-β is overexpressed [31], and consequently, it is more available and active, causing a series of harmful reactions, such as inflammation and tissue fibrosis and activating matrix metalloproteinases (MMPs), in particular MMP-2 and MMP-9 [30,32]. These enzymes are responsible for the degradation of structural components of the extracellular matrix such as collagen and elastin that contribute to the strength and flexibility of the aortic wall, which progressively weakens the aortic wall until dilation and dissection [31].

Genetic alterations are therefore responsible for the histological features of MFS, which is characterized by areas of cystic degeneration of the aortic media caused by translamellar MEMA, with loss of smooth muscle cells and fragmentation of elastic fibres [20].

Table 2 summarizes the main characteristics.

### 3.2. Loeys–Dietz Syndrome

Loeys–Dietz syndrome (LDS) is a genetic connective tissue disease with broad-spectrum multisystem involvement [33,34].

LDS is characterized by craniofacial, skeletal, cutaneous, and cardiovascular abnormalities, which are responsible for mortality and morbidity at a young age [35].

Cardiovascular manifestations affect the valves and arteries. At the valvular level, LDS presents with dilation of the valvular ring and mitral valve prolapse. At the aortic level, it manifests with aneurysm of the aortic root with a high risk of aortic dissection and consequently of sudden cardiac death in young people; in addition, it is common to observe an abnormal tortuosity of the arteries, especially the carotid and vertebral arteries [36,37].

Other less frequent cardiac manifestations of LDS are cardiomyopathy, atrial fibrillation, patent ductus arteriosus, and atrial septal defects [35].

LDS is a rare disease [34,38], whose incidence and prevalence are unknown; it can occur in both sexes and in different ethnic groups.

LDS is transmitted in an autosomal dominant manner, with two thirds of cases resulting from de novo mutations associated with more severe phenotypes and one third with familial origin associated with milder phenotype; cases of non-penetrance and mosaicism are also reported [33].

LDS is caused by pathogenic variants in genes encoding TGF-β receptors and other proteins involved in the TGF-β pathway, resulting in dysfunction in TGF-β signalling, which alters the normal development and maintenance of connective tissue.

The main mutated genes are *TGFBR1* and *TGFBR2* encoding TGF-β receptor type 1 and 2, *SMAD3* encoding a protein that mediates intracellular TGF-β signalling, and *TGFB2* and *TGFB3* encoding TGF-β isoforms 2 and 3.

Five major subtypes have been identified based on gene mutations [34].

LDS Type 1 is associated with pathogenic variants in the *TGFBR1* gene, is the most severe phenotype, and is characterized by significant craniofacial abnormalities. LDS Type 2 is associated with pathogenic variants in the *TGFBR2* gene and is characterized by minimal craniofacial abnormalities compared to type 1. LDS Type 3 is associated with pathogenic variants in the *SMAD3* gene and is characterized by osteoarthritis. LDS Type 4 is associated with pathogenic variants in the *TGFB2* gene. LDS Type 5 is associated with pathogenic variants in the *TGFB3* gene and is the least severe phenotype.

Although pathogenic variants in *SMAD2* have been identified, they have not yet been placed within LDS subtypes [39].

Histologically, SLD is characterized by areas of cystic degeneration of the tunica media of the aorta caused by MEMA predominantly of the intralamellar type, with loss of smooth muscle cells, fragmentation, and disorganization of elastic fibres [20].

Table 2 summarizes the main characteristics.

### 3.3. Vascular Ehlers–Danlos Syndrome

Vascular Ehlers–Danlos syndrome (VEDS) is a rare inherited disorder characterized by fragile connective tissues caused by abnormalities in type III collagen [40].

VEDS is part of Ehlers–Danlos syndrome (EDS), a heterogeneous group of connective tissue disorders characterized by skin fragility, joint hyperlaxity, and vascular fragility.

The 2017 diagnostic criteria describe 13 different types of EDS and indicate when genetic testing is recommended. Testing is recommended when one of the following major criteria is present: positive family history, arterial rupture or dissection before age 40, spontaneous perforation of the sigmoid colon, unexplained uterine rupture in the third trimester, or spontaneous formation of a carotid-cavernous fistula. Testing is also recommended when multiple minor criteria are present, such as characteristic appearance, spontaneous pneumothorax, equinus foot, congenital hip dislocation, hypermobility, keratoconus, and tendon or muscle rupture [41].

VEDS is the most severe form of EDS, with a high risk of cardiovascular complications. Cardiovascular manifestations mainly affect the mitral valve with valvular insufficiency due to tissue laxity, the aorta with aneurysm, aortic dissection and rupture often with multiple and simultaneous sites [42] up to sudden cardiac death, and small–medium-calibre vessels prone to rupture at a young age [29].

Spontaneous arterial dissections, intestinal rupture, and uterine rupture are the most characteristic and life-threatening complications of VEDS [43].

VEDS is extremely rare, with an estimated prevalence of 1/50,000 to 200,000 people, accounting for 5% of all EDS patients, with no gender predominance [42,44], and sudden death may occur before the age of 20 and is more common in males [45].

VEDS is transmitted in an autosomally dominant manner with high penetrance and is caused by pathogenic variants in the *COL3A1* gene, located on chromosome 2q32.2 and consisting of 51 coding exons that encode a protein of 1466 amino acids [46].

In half of the cases, these variants are missense substitutions of a glycine in the repeat sequence (Gly-X-Y) within the triple-helix region of type III collagen [43], a structural protein essential for the integrity of connective tissues, particularly in blood vessels, skin, and internal organs.

There are rarely pathogenic variants of *COL3A1* that cause haploinsufficiency with milder phenotypes [47] and in less than 1% bi-allelic variants [48], and furthermore, heterozygous arginine–cysteine substitutions in *COL1A1* rarely cause a phenotype like VEDS [43].

VEDS on histology is characterized by areas of cystic degeneration of the tunica media of the aorta caused by translamellar-type MEMA [20].

Table 2 summarizes the main characteristics.

## 4. Non-Syndromic Hereditary Aortopathies

### 4.1. Non-Syndromic Familial Thoracic Aortic Aneurysm and Dissection

Non-syndromic thoracic aortic aneurysm and dissection (NS-TAAD) is a rare genetic disease, usually clinically silent, but often fatal until aortic dissection or rupture [49] and subsequent sudden cardiac death.

NS-TAAD can be classified into familial NS-TAAD when more than one family member is affected and sporadic NS-TAAD when no other family members are affected [50].

Familial NS-TAAD is sometimes discovered incidentally, but often the first obvious sign is sudden death or acute aortic dissection affecting a young member of the family.

Familial NS-TAAD is predominantly inherited in an autosomally dominant manner with reduced penetrance and variable expressivity, and the genetic aetiology is highly heterogeneous and only 20% can be attributed to pathogenic variants [51,52,53]; therefore, diagnosis is very difficult.

The ClinGen Aortopathy Expert Panel has classified five genes that mainly cause isolated NS-TAAD [54]. These genes include *ACTA2*, *MYH11*, *MYLK*, and *PRKG1*, which encode components of the contractile apparatus of vascular smooth muscle cells (SMCs), and *LOX*, encoding an extracellular enzyme involved in collagen and elastin cross-linking [55,56].

The major familial NS-TAAD gene is *ACTA2*, responsible for 14% to 21% of cases, encoding aortic α-smooth muscle actin and a target for TGF-β signalling, and the known mutations are all missense or in-frame indels. Despite a penetrance of approximately 50%, patients may develop other vascular abnormalities and smooth muscle cell functional disorders, such as cerebral aneurysm, Moyamoya-like neurovascular malformations, coronary artery disease, and livedo reticularis [51,57].

Other genes account for ≤1% of NS-TAAD [51].

The *MYH11* gene, encoding the SM myosin heavy chain, has genetic variants of unknown significance (VUS), but known pathogenic variants are mainly missense or in-frame indels, interfering with protein interactions, and with incomplete penetrance [51,57].

The *MYLK* gene produces three isoforms, which can cause haploinsufficiency or destruction of the CaM-binding domain; these patients are difficult to manage as aortic dilation is often not observed before dissection [58].

The *PRKG1* gene has a recurrent p.Arg177Gln gain-of-function mutation, which causes TAAD with complete penetrance, inhibiting cGMP binding and being hyperactive even in its absence [59].

The *LOX* gene, encoding lysyl oxidase, is essential for extracellular matrix maturation. Heterozygous loss-of-function mutations, which impair catalytic activity or lead to haploinsufficiency, predispose to TAAD, especially in the root and ascending aorta [60].

On histology, NS-TAAD presents medial degeneration and fragmentation and loss of elastic fibres [20].

Table 3 summarizes the main characteristics.

### 4.2. Non-Syndromic Bicuspid Aortic Valve

Bicuspid aortic valve (BAV) is a congenital heart defect of the aortic valve characterized by the presence of two cusps instead of three, which leads to degenerative changes in the valve and is associated with aortopathy [61,62].

BAV can be divided into syndromic BAV (S-BAV) when it is associated with genetic syndromes and non-syndromic BAV (NS-BAV) when it has isolated gene mutation.

BAV causes valvular alterations such as valvular incompetence, valvular stenosis due to dystrophic calcification, infective endocarditis [63], and is associated with aortopathies such as ascending aortic dilation from 40% to 60%, coarctation of the aorta in 50–75%, aortic aneurysm [61,62], aortic dissection or aortic rupture, and, consequently, sudden cardiac death [63,64].

BAV is one of the most frequent congenital cardiac anomalies with a prevalence between 0.5% and 0.77% [65] and has a prevalence of 30% in Turner syndrome and William syndrome [61] and may be associated with other syndromes such as Loeys–Dietz syndrome, velocardiofacial syndrome, and occasionally Down syndrome, Alagille syndrome, and Kabuki syndrome [66].

In addition, BAV may be associated with other congenital heart defects such as ventricular septal defects (VSDs) with aortic arch obstruction in 51%, adult aortic arch coarctations in 37%, isolated VSDs in 20.5%, AV canals in 7.5%, Tetralogy of Fallot in 2%, and complete TGA in 1% [64].

Familiality occurs in approximately 9% of BAV cases [64], and NS-BAV is inherited as an autosomally dominant trait with incomplete penetrance and variable expressivity [66], and according to the 2022 AHA/ACC guidelines, BV meets the criteria for screening first-degree relatives [4].

Current single-gene mutations explain less than 10% of BAV cases, and the major mutated genes associated with NS-BAV and aortic dilatation (TAA) are *NOTCH1*, *GATA4–6*, *SMAD4* and *SMAD6*, *ROBO4*, *ACTA2*, and *FBN1* [66].

*NOTCH1* is the most frequently responsible genetic variant for NS-BAV, encoding a receptor involved in endothelial–mesenchymal transition and cardiac valve development [66]. *NOTCH1* loss-of-function mutations can accelerate aortic valve calcification in BAV [67]. Rare variants in other Notch pathway genes such as *ARHGAP31*, *MAML1*, *SMARCA4*, *JARID2*, and *JAG1* are associated with BAV and aortic coarctation [68].

*GATA4–6* encode zinc finger transcription factors that regulate early cardiac gene expression and cardiac cell lineage differentiation [69]: specifically, common variants of *GATA4* are associated with BAV, and rare variants of *GATA4*, *GATA5*, and *GATA6* have been identified in studies [66].

Variants of *SMAD4* and *SMAD6*, proteins that transduce TGF-β signals, have been found in NS-BAV, and rare variants of *ROBO4* expressed in endothelial cells have been identified in NS-BAV [66].

Pathogenic variants in *ACTA2*, which encodes smooth muscle alpha-actin, are relatively rare in NS-BAV cases [70]. Rare variants of *FBN1* encoding an extracellular glycoprotein (fibrillin-1) have been found in NS-BAV and aortic root aneurysms [66].

Furthermore, many of these pathogenic variants are also associated with other pathological conditions.

Heterozygous pathogenic variants of the *NOTCH1* gene are associated with congenital heart defects, both left- and right-sided, characterized by incomplete penetrance and phenotypic variability [71].

Pathogenic variants of the transcription factors *GATA4*, *GATA5*, and *GATA6* are responsible for severe congenital heart defects, such as double-outlet right ventricle (DORV) and ventricular septal defects (VSDs) [72].

*SMAD4* is a central mediator that regulates cellular processes such as growth, apoptosis, and migration, affecting tumour initiation and progression. Pathogenic *SMAD4* variants are associated with various diseases, such as cancer and juvenile polyposis syndrome (JPS).

In cancers, the loss or inactivation of *SMAD4* impairs the ability of TGF-β to suppress tumours, instead promoting cancer progression. In particular, *SMAD4* is frequently mutated in pancreatic cancer, colorectal cancer, and juvenile polyposis, where its dysfunction leads to the formation of gastrointestinal polyps and increases the risk of malignancies [73].

In JPS, pathogenic *SMAD4* variants are associated with an increased risk of gastric polyps and extraintestinal manifestations such as hereditary haemorrhagic telangiectasia (HHT), which can cause vascular malformations and bleeding [74].

*SMAD6* is a gene that encodes a protein belonging to the SMAD family, which functions as an inhibitor of the Bone Morphogenetic Protein (BMP) signalling pathway. The BMP pathway regulates cell proliferation and differentiation, which is essential for the development of different tissues such as bone, heart, muscle, and nervous system tissue.

*SMAD6* functions to block the activation of this pathway, preventing the interaction between BMP receptors and SMAD1/5/8 proteins, which are necessary to transmit the signal in cells. However, pathogenic mutations of *SMAD6* can reverse this function, turning SMAD6 from an inhibitor to an activator of the BMP pathway.

Pathogenic *SMAD6* variants are associated with birth defects, including congenital heart disease and craniosynostosis [75].

On histological examination, BAV is often characterized by fibrosis and calcification of the cusps, the endothelium lining the valve may be thickened and irregular, with an abnormal distribution of the valve interstitial cells. The associated aortic wall may show severe non-inflammatory degenerative changes such as elastic fibre fragmentation, smooth muscle cell death, and MEMA [64].

Table 3 summarizes the main characteristics.

## 5. Molecular Autopsy and Prevention Strategies

SCD is one of the leading causes of mortality worldwide and, particularly in young people, is often associated with undiagnosed cardiac conditions of genetic origin. In such cases, molecular autopsy (MA), defined as post-mortem genetic analysis, is crucial for identifying any pathogenic variants responsible for hereditary conditions [7]. However, despite the use of MA, in 5–10% of cases, the cause of SCD remains unknown and is classified as Sudden Unexplained Death (SUD) [7]. In 10–20% of SUD cases, a pathogenic genetic mutation is found [76].

In the young population, the primary causes of sudden cardiac death in the young (SCDY) are channelopathies, cardiomyopathies, and hereditary aortopathies [6]. When a specific genetic mutation associated with a particular phenotype is identified, these are referred to as syndromic aortopathies, such as Marfan syndrome (MFS), Loeys–Dietz syndrome (LDS), and vascular Ehlers–Danlos syndrome (VEDS). Conversely, when aortic manifestations are caused by pathogenic variants affecting the components of vascular smooth muscle cells but are not linked to other systemic abnormalities, they are termed non-syndromic aortopathies [29].

In 2015, the American College of Medical Genetics and Genomics (ACMG) and the Association for Molecular Pathology (AMP) issued guidelines for the evaluation of genetic sequence variations, establishing five categories of variant classification to ensure consistency across laboratories [77]. They recommend using standard terminology to classify variants as pathogenic, likely pathogenic, likely benign, benign, or of uncertain significance (VUS) [77,78,79,80].

The classification process involves assessing population data, functional data, and computational analyses. Identifying a pathogenic variant requires a multidisciplinary approach that combines bioinformatics tools, functional and clinical data, segregation analysis, and consultation of genetic databases. A key criterion in evaluating a variant is its frequency in the general population. A variant is considered benign if it is too common to be associated with a rare disease, whereas a rare variant may be suspected of having a pathogenic effect in the context of Mendelian genetic disorders.

In silico tools are employed to predict the functional impact of variations. Bioinformatic programs such as PolyPhen-2 (v2.2.2r398), SIFT (version 6.2.1), and MutationTaster 2 assess whether a missense mutation is likely to impair protein function. PolyPhen-2 predicts the probability that an amino acid substitution will damage protein function, while SIFT evaluates whether an amino acid change is tolerated or potentially harmful.

Functional studies, such as enzymatic assays or protein expression experiments, can directly demonstrate a variant’s impact on biological function. A variant is considered pathogenic if it leads to a loss of function or malfunction of the encoded protein. Segregation analysis also provides evidence of pathogenicity: if a variant is present in all affected family members but absent in healthy ones, this strongly suggests it is pathogenic.

Furthermore, variants that cause nonsense or frameshift “null” mutations in genes where loss of function is known to cause disease are deemed pathogenic. Similarly, if a variant produces the same amino acid change as a previously recognised pathogenic mutation, it is considered pathogenic. To confirm whether a variant has been identified as pathogenic or is associated with specific diseases, comparison with databases such as ClinVar, OMIM, and HGMD and reviewing relevant scientific literature are essential.

Lastly, the variant must be linked to the patient’s clinical phenotype. If the variant is in a gene known to cause a disease associated with the patient’s symptoms, it is more likely to be pathogenic.

One of the most challenging aspects of MA is interpreting VUS—genetic mutations that are neither clearly pathogenic nor benign. In clinical practice, VUSs pose a challenge because, without conclusive data on their pathogenicity, it is difficult to determine whether they are responsible for the condition that caused death.

Recognising a hereditary or genetic component is critical not only for understanding the cause of the patient’s death but also for preventing further deaths in the victim’s family [73]. This can initiate screening and clinical monitoring programmes to mitigate the risk of future SCD.

Once the cause of SCDY is identified, it is crucial to implement a multifactorial preventive strategy for the victim’s relatives, aimed at reducing their risk. The first step in preventing SCDY involves a thorough clinical evaluation of the victim’s family members, including cardiological screening and genetic testing. Cardiological screening may include an electrocardiogram, echocardiogram, and, when indicated, Holter monitoring and stress testing to identify electrical or structural abnormalities that could predispose them to cardiac arrest. Genetic testing is vital when SCD is suspected to have a genetic origin, as it helps identify specific mutations in family members, enabling continuous monitoring and preventive treatments.

The second line of intervention should involve psychological support and genetic counselling. Adequate psychological support is essential, as sudden loss can lead to post-traumatic stress, anxiety, or depression in family members. Support helps them manage the emotional impact and alleviate concerns about their own heart health. A consultation with a geneticist before and after genetic testing is also valuable, as it provides an opportunity to discuss the implications of the results and the potential therapeutic and surveillance options available to the family.

Preventing SCDY in the families of victims requires a multidisciplinary approach involving experts, with the strategy tailored to the individual and familial risk profile.

## 6. Conclusions and Future Perspectives

HAs represent a heterogeneous group of genetic pathologies, which can affect the aorta and other cardiovascular structures, leading to serious complications such as dilation, dissection, aortic rupture, and sudden cardiac death.

Early diagnosis and timely management of these pathologies are crucial to improve patients’ life expectancy. Developments in molecular genetics, such as the identification of pathogenic variants in the *FBN1*, *TGFBR*, *COL3A1*, and other genes, have enabled a greater understanding of the pathogenetic mechanisms underlying these diseases. However, a significant portion of the pathogenic variants associated with such aortopathies is still unknown, highlighting the importance of continued research in this field.

MA and genetic prevention represent key tools for early identification and management of patients at risk, especially in families with a history of SCD. Early diagnosis in victims’ family members through genetic testing can allow targeted preventive interventions and improve the prognosis of affected patients, reducing the incidence of fatal events.

## Figures and Tables

**Table 1 diseases-12-00264-t001:** Hereditary aortopathies classification.

Hereditary Aortopathy	Pathogenic Variants	Cardiac Alterations	Cardiac Complications
** Syndromic hereditary aortopathies **			
Marfan Syndrome	FBN1 (90%) TGFBR1 TGFBR2	Mitral and tricuspid valve prolapse with valvular insufficiency and aortic dilation up to aneurysm	Dissection and rupture of the aorta and sudden cardiac death
Loeys–Dietz Syndrome	*TGFBR1* *TGFBR2* *SMAD2* *SMAD3* *TGFB2* *TGFB3*	Dilatation of the aortic root at the sinuses of Valsalva	Dissection and rupture of the aorta and sudden cardiac death
Vascular Ehlers–Danlos syndrome	COL3A1	Mitral valve with valvular insufficiency, dilation and aortic aneurysm, and rupture of small- and medium-calibre vessels	Dissection and rupture of the aorta and sudden cardiac death
** Non-Syndromic ** **hereditary aortopathies**			
Non-syndromic familial thoracic aortic aneurysm and dissection	ACTA2 MYH11 MYLK PRKG1 LOX	Thoracic aortic aneurysm	Dissection and rupture of the aorta and sudden cardiac death
Non-syndromic bicuspid aortic valve	NOTCH1 GATA4–6 SMAD4 SMAD6 ROBO4 ACTA2 FBN1	Ascending aortic dilation, coarctation of the aorta, and aortic aneurysm	Dissection and rupture of the aorta and sudden cardiac death

**Table 2 diseases-12-00264-t002:** Syndromic hereditary aortopathies.

Syndrome	Epidemiology	Hereditary Transmission	Pathogenic Variants	Cardiovascular Manifestations	Histological Features
Marfan syndrome (MFS)	Most frequent monogenic malformation syndrome, incidence 1:3000–1:5000, without differences between sex or ethnic groups; the average life expectancy of patients is about 40 years	Autosomally dominant, de novo mutations in 25% of cases, high penetrance	FBN1 (90%), encoding fibrillin-1, TGFBR1, TGFBR2	Mitral prolapse, mitral insufficiency, ascending aortic dilation, aortic aneurysm, aortic dissection, aortic insufficiency, sudden cardiac death	Areas of cystic degeneration of the aortic media caused by translamellar MEMA, with loss of smooth muscle cells and fragmentation of elastic fibres
Loeys-Dietz syndrome (LDS)	Rare disease, incidence and prevalence are unknown, it can occur in both sexes and in different ethnic groups	Autosomally dominant, with frequent de novo mutations	*TGFBR1*, *TGFBR2*, *SMAD2*, *SMAD3*, *TGFB2*, *TGFB3*	Mitral prolapse, aortic root aneurysm, aortic dissection, arterial tortuosity, cardiomyopathy, atrial defects, sudden cardiac death	Areas of cystic degeneration of the tunica media of the aorta caused by MEMA predominantly of the intralamellar type, with loss of smooth muscle cells, fragmentation and disorganization of elastic fibres
Vascular Ehlers-Danlos syndrome (VEDS)	Extremely rare, prevalence 1/50,000 to 200,000, no gender predominance, sudden death may occur before the age of 20 and is more common in males	Autosomally dominant, high penetrance	COL3A1, encoding type III collagen, COL1A1 rare and causes similar phenotypes	Mitral regurgitation, aortic aneurysm, aortic dissection and rupture, spontaneous arterial rupture, sudden cardiac death	Areas of cystic degeneration of the tunica media of the aorta caused by translamellar-type MEMA

**Table 3 diseases-12-00264-t003:** Non-syndromic hereditary aortopathies.

Syndrome	Epidemiology	Hereditary Transmission	Pathogenic Variants	Cardiovascular Manifestations	Histological Features
Non-syndromic thoracic aortic aneurysm and dissection (NS-TAAD)	Rare, incidence unknown	Autosomally dominant, variable penetrance	ACTA2, MYH11, MYLK, PRKG1, LOX	Aortic dilation, aortic dissection, aortic rupture, sudden cardiac death	Medial degeneration and fragmentation and loss of elastic fibres
Bicuspid aortic valve (BAV)	Most frequent congenital cardiac anomalies, prevalence between 0.5% and 0.77%	Autosomally dominant, incomplete penetrance, variable expressivity	NOTCH1, GATA4–6, SMAD4, SMAD6, ROBO4, ACTA2, FBN1	Valvular incompetence, valvular stenosis, infective endocarditis, ascending aortic dilation, coarctation of the aorta, aortic aneurysm, aortic dissection, aortic rupture, sudden cardiac death	Fibrosis and calcification of the cusps, the endothelium lining the valve may be thickened and irregular, with an abnormal distribution of the valve interstitial cells

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
