# Peer review of "Hereditary Aortopathies as Cause of Sudden Cardiac Death in the Young: State-of-the-Art Review in Molecular Medicine"

_diseases, 2024, doi:10.3390/diseases12110264_

Round 1
Reviewer 1 Report
Comments and Suggestions for Authors
Authord have addressed an interesting issue and in-depth described Hereditary Aortopathies.
The manuscript is well-organized.
I suggest to better describe strategies to unreveal pathogenic gene mutations. They introduced this issue but it deserves to be more in-depth discussed (i.e. how to indentify a pathogenetic variant?).
Next, some of the gene indicated (i.e. SMAD4) are involved in different diseases. This should be more in-depth discussed.
As a minor point, which is the mean of MCI? Stay it for SCD?
Author Response
Dear Reviewer1,
Thank you for your comments that improve the review article, as indicated:
Comments 1: I suggest to better describe strategies to unreveal pathogenic gene mutations. They introduced this issue but it deserves to be more in-depth discussed (i.e. how to indentify a pathogenetic variant?).
Response 1: We described the strategies to detect pathogenic genetic mutations according to the guidelines of the American College of Medical Genetics and Genomics and the Association for Molecular Pathology [74] (lines 416-441 and highlighted in green).
Comments 2: Next, some of the gene indicated (i.e. SMAD4) are involved in different diseases. This should be more in-depth discussed.
Response 2: We added a discussion of genes responsible for other diseases with new bibliographic references: line 355-384, highlighted in light blue, references 72 to 76.
Comments 3: As a minor point, which is the mean of MCI? Stay it for SCD?
Response 3: We corrected MCI to SCD in the text (lines 395 and 399 and highlighted in yellow).
In addition, we replaced “mutation” with “pathogenic variant” according to the guidelines of the American College of Medical Genetics and Genomics and the Association for Molecular Pathology [74].
Sincerely
Reviewer 2 Report
Comments and Suggestions for Authors
This review systematically summarized the pathogenesis, histopathology, classification of aortopathies and respectively introduced the genetic basis underlying syndromic and non-syndromic aortopathies. Generally, this manuscript is informative and well-organized. The following concerns may be addressed to further improve the readability.
1. Writing/syntax needs extensive editing. for example, in the Abstract section, the sentence "The post-mortem genetic study defined as molecular autopsy (MA) is essential for HA may be useful to confirm suspected aortopathy discovered at autopsy" is very confusing.
2. Some abbreviations should be written with their full name at their first mention. For example, SCDY in the Abstract section should has its full name.
3. In the Introduction section, authors may make comments on a recent publication regarding sudden unexplained death in the young (PMID: 37624372). This publication have identified importantly some novel genes as substrates of sudden unexplained death in the young and thus increased the genetic testing yield.
4. Authors well described the common gene mutations associated with aortopathies. I suggest authors add some tables to more quickly convey their conclusions in the forth section.
Comments on the Quality of English Language
Writing and syntax should be extensively improved.
Author Response
Dear Reviewer2,
Thank you for your comments that improve the review article, as indicated:
Comments 1: Writing/syntax needs extensive editing. for example, in the Abstract section, the sentence "The post-mortem genetic study defined as molecular autopsy (MA) is essential for HA may be useful to confirm suspected aortopathy discovered at autopsy" is very confusing.
Response 1: We have corrected the sentence indicated in the abstract to “When the cause of death remains unknown after an autopsy, it is referred to as sudden unexplained death (SUD), and postmortem genetic testing, known as a molecular autopsy (MA), is crucial to confirm HA and assess the genetic risk in the patient’s relatives. This helps to facilitate diagnostic and therapeutic pathways and/or implement monitoring strategies to prevent SCD.” (line 18-22 and highlighted in yellow)
Comments 2: Some abbreviations should be written with their full name at their first mention. For example, SCDY in the Abstract section should has its full name.
Response 2: We have written the abbreviations with their full name at the first mention and have always written them with their full name in the abstract.
Comments 3: In the Introduction section, authors may make comments on a recent publication regarding sudden unexplained death in the young (PMID: 37624372). This publication have identified importantly some novel genes as substrates of sudden unexplained death in the young and thus increased the genetic testing yield.
Response 3: We have added a discussion of the indicated article in the introduction [15] Wang, S., Chen, Y., Du, J., Wang, Z., Lin, Z., Hong, G., Qu, D., Shen, Y., & Li, L. (2023). Postmortem genetic analysis of sudden unexplained death in a young cohort: a whole-exome sequencing study. International journal of legal medicine, 137(6), 1661–1670. (line 51-63 and highlighted in green).
Comments 4: Authors well described the common gene mutations associated with aortopathies. I suggest authors add some tables to more quickly convey their conclusions in the forth section.
Response 4: We added 2 summary tables of syndromic and non-syndromic aortopathies.
Additionally, we reviewed the English with artificial intelligence and then an expert reviewed the English for syntax and eventual repetition.
Sincerely.
Round 2
Reviewer 2 Report
Comments and Suggestions for Authors
Authors have sufficiently addressed my concerns.